# Grounding and Validation of Algorithmic Recourse in Real-World Contexts: A Systematized Literature Review

## Abstract

The aim of algorithmic recourse (AR) is generally understood to be the provision of "actionable" recommendations to individuals affected by algorithmic decision-making systems, in an attempt to offer the capacity for taking actions that may lead to more desirable outcomes in the future. Over the past few years, AR literature has largely focused on theoretical frameworks to generate "actionable" counterfactual explanations that further satisfy various desiderata, such as diversity or robustness. We believe that algorithmic recourse, by its nature, should be seen as a practical problem: real-world socio-technical decision-making systems are complex dynamic entities involving various actors (end users, domain experts, civil servants, system owners, etc.) engaged in social and technical processes. Thus, research needs to account for the specificities of systems where it would be applied. To evaluate how authors envision AR "in the wild", we carry out a systematized review of 127 publications pertaining to the problem and identify the real-world considerations that motivate them. Among others, we look at the ways to make recourse (individually) actionable, the involved stakeholders, the perceived challenges, and the availability of practitioner-friendly open-source codebases. We find that there is a strong disconnect between the existing research and the practical requirements for AR. Most importantly, the grounding and validation of algorithmic recourse in real-world contexts remain underexplored. As an attempt to bridge this gap, we provide other authors with five recommendations to make future solutions easier to adapt to their potential real-world applications.

## 1 Introduction

Algorithmic decision-making (ADM) tools are frequently seen as a way to improve decision processes in a variety of high-stakes domains such as public administration [47, 146] or healthcare [45, 87]. Deep learning models have attracted much attention due to their perceived high performance, but the predictions of such models cannot be interpreted by humans, hence end users – both individuals subjected to algorithmic decisions and decision-makers operating on them – are placed in a position where they are unable to understand the grounds of a prediction, act on it, or trust it [159].

To help address this problem, a variety of explanation methods has been proposed. Of particular interest for this paper are counterfactual explanations (CEs) that attempt to explain the predictions for individual instances of data, taking the form of conditional statements such as *"if the value of feature $x$ was $a$ instead of $b$, the model would have predicted class $y$ instead of $z$"*. They are perceived to be an attractive approach to explanation that does not require "opening the black box" [151] and have been argued to align with the ways that humans naturally reason about events [84].

Submitted to 38th Conference on Neural Information Processing Systems (NeurIPS 2024). Do not distribute.

CEs are also seen as the go-to method for algorithmic recourse (AR), or the generation of actionable recommendations that provide people with the knowledge needed to achieve more desirable predictions in ADM systems. Recourse is distinct from the "explanation" or "justification" of algorithmic decisions, and more closely related to the notion of contestability of Artificial Intelligence [7] in that it aims not only to improve the *trust* in the algorithm, but also embrace human *agency* [142].

Algorithmic recourse is an inherently practical problem in that it resembles a bureaucratic complaint process: an individual unhappy with some decision engages with a representative of the issuing organization, in an attempt to overturn it. Yet, we observe that much of the existing work is highly theoretical, with little consideration of whether it could be applied in organizational settings [see also 18]. Deploying AR in realistic systems without analyzing its mechanics in a broader context and without knowing what types of dynamics are expected to arise is bound to lead to unanticipated outcomes. Many of them will be undesirable and even potentially unsafe, and impossible to validate with respect to a set of requirements because the requirements for AR are *necessarily* socio-technical.

**Societal and institutional components of algorithmic recourse are the focal point of our work**, as we look beyond the typical technical considerations to assess the practical aspects of the problem.

To that end, we contribute a *systematized review* of 127 publications that address the goals of algorithmic recourse and we evaluate to what extent they incorporate such practical considerations. We characterize our approach as *systematized* because we follow a fully systematic approach to the collection of publications, but their selection is not necessarily exhaustive [46] as many impactful ideas in computer science are published only in the form of pre-prints. Based on our analysis, we also provide other authors with five recommendations on how to improve the practicality of AR research.

The rest of the manuscript is structured as follows. In Section 2 we elaborate on the background of our work. Then, in Section 3 we describe our approach to this review. Next, Section 4 introduces our findings. Section 5 provides a discussion of our results, introduces our recommendations, and addresses the limitations of the current work. Finally, Section 6 forms the conclusion to this paper.

## 2 Background

### 2.1 On algorithmic recourse

Algorithmic – or actionable, individual – recourse was introduced in [138] as *"the ability of a person to change the decision of the model through actionable input variables"*, building on the earlier work of [151] who argued that CEs are a psychologically-grounded way to (1) help decision-subjects understand an algorithmic decision, (2) provide them with information needed to contest it, and (3) inform about actions that could be taken to overturn it. For instance, consider a person who has unsuccessfully applied for a loan; they may then receive AR such as *"if you requested $5000 less, you would qualify for this loan"*. The key consideration for AR is "actionability", which entails that the recipient of the recommendation should be capable of implementing it. If they had been informed *"if you were 10 years younger, you would qualify for the loan"*, they would have still received a valid CE, *but not* recourse. More recently [69] has recast the problem as reasoning about minimal interventions on the structural causal model. This formulation (at least theoretically) addresses an important shortcoming of "correlational" recourse. Without accounting for the downstream causal effects of actions, an individual may exert more effort than necessary and still fail to achieve the target outcome. Indeed, counterfactuals are an inherently causal concept [103].

We note that problems similar to AR have been studied under a variety of different names: *actionable knowledge discovery* [e.g., 2], *action rules mining* [e.g., 110], *inverse classification* [e.g., 5], *why not questions* [e.g., 58], or *actionable feature tweaking* [134]. These alternative formulations have generally focused on "business" knowledge, rather than individual recommendations, but ultimately the goal of all these approaches is to extract information from a (black-box) model that allows the user – an individual or a decision-maker – to act. We highlight them to emphasize that AR does not have to be achieved through the means of CEs. Rather CEs should be seen as *one of the means* to achieve AR, particularly promising in that they do not require expert-level understanding of the model to be useful. Nonetheless, we decide to distinguish between the literature on AR (commonly equated with actionable CEs), and these alternative formulations in our work.

Existing research has generally considered AR in simplistic settings that are far removed from real-world socio-technical decision-making systems, where it would be implemented as a process.

For example, such systems are dynamic [113, 137], must support the implementation of AR at scale [9, 94], and involve various stakeholders beyond the end users [17, 151]. Moreover, if the intended goal of AR is to help individuals subjected to algorithmic decisions in an effective manner, research must entail a rich understanding of "actionability" to account for the differences between them [142].

## 2.2 On the position of our review

Several groups of authors have previously surveyed the landscape of counterfactual explanations in general, and algorithmic recourse specifically. Perhaps the most relevant to our work is [71], which discusses five deficits of research on CEs, with a special focus on the (lack of) psychological grounding. Another pertinent publication is [70], which attempts to unify the definitions and formulations of AR in existing literature, but the work primarily focuses on technical aspects. Next, [143] develops a rubric to compare counterfactual explainers (equated with AR) and identifies 21 research challenges. While these also remain mostly technical, several of them are relevant to our work, for instance, CEs *"as an interactive service to the applicants"* or reinforcing *"the ties between machine learning and regulatory communities"*. More recently, [48] reviewed and benchmarked a number of CE generators, but AR is only a secondary consideration in the work. We also highlight [130], which is the only systematic review of counterfactual and contrastive approaches to date. The authors understand CEs as a way to justify model predictions (i.e., they are different from AR). We agree with this distinction in that CEs can be useful for reasons other than recourse, such as model debugging [e.g., 1, 122]. Finally, although not reviews, [13] and [142] are particularly relevant to our work, offering critical perspectives on AR and addressing multiple shortcomings of recourse literature.

## 3 Methods

In this section, we briefly discuss our approach to the literature review following the SALSA – Search, Appraisal, Synthesis, Analysis – framework introduced in [46]. We also provide a more detailed description to allow for the reproduction of our process in the supplementary materials. Figure 1 presents our process in the form of a PRISMA flow diagram [97].

## 3.1 Search

We make use of three search engines to collect the initial set of studies: ACM Digital Library, IEEE Xplore, and SCOPUS. Given the previously mentioned blurry distinction between AR and CEs, we consider the papers discussing either problem. In a small scoping review, we identify several keywords common to publications on recourse, as well as several equivalent terms to build the query. We search in titles, abstracts, and keywords, arriving at 3092 records after de-duplication. To facilitate the screening process, we employ the open-source ASReview tool, which makes use of an active learning approach to re-order the set of publications, such that the most relevant ones are always "at the top of the stack" [139]. The researchers behind the tool suggest employing a stopping rule measured in the number of consecutive irrelevant records, which we set to 30, or 1% of the entire dataset. We accept all papers that focus on algorithmic recourse and counterfactual explanations, completing the screening after evaluating 1040 abstracts, leading to 499 relevant records.

We observe that some important publications may be missing from our results. For instance, [151] was published in a legal journal that is not indexed by computer science search engines. Thus, we decide to augment the set of records by applying snowballing, which has been shown as a good alternative to databases in systematic reviews in software engineering [162]. We collect the references for the top 50 (10%) "most impactful" publications, measured by the number of citations. While this introduces several pre-prints into our result set [52, 61, 91, 113, 143, 150], we decide not to exclude them. Our review remains primarily concerned with peer-reviewed work. After adding the snowballed references to our dataset, we are left with 2018 records for the second screening with ASReview. This time, we look for publications that specifically refer to the problem of AR, "actionable" CEs, or modifying outcomes of automated decision-making systems. We employ a stricter stopping rule to minimize the risk of false negatives, completing the screening after 60 consecutive irrelevant records with 203 records considered for full-text appraisal. To allow for complete reproducibility of the search process, we provide an extended discussion (including queries) in the technical Appendix A.

## 3.2 Appraisal

We were able to retrieve all of the remaining 203 documents. For each document, we require that the authors explicitly cite recourse as the center of interest, or look at **(1)** explanations **(2)** provided for individual instances **(3)** with the goal of acting upon them **(4)** in an attempt to modify the predictions **(5)** of a classification model. We exclude 51 publications as they are not on topic, primarily because they focus on CEs for the sake of explanation. Four works in this category look at (what they call) recourse but extend the problem to settings beyond the scope of this review: recommender systems [31, 43, 145], text classification [37], and anomaly detection [27]. Further 15 publications are duplicates, typically pre-prints of other documents that were included in the review. Next, 8 documents were published before [151] that sparked the research on AR, and thus we exclude them as well. These look at the alternative formulations discussed earlier in Section 2.1. Finally, 2 documents are not publications: one is an abstract of a talk, and the other is a student poster. For each document, we answer a number of questions relating to the practical considerations introduced by the authors.

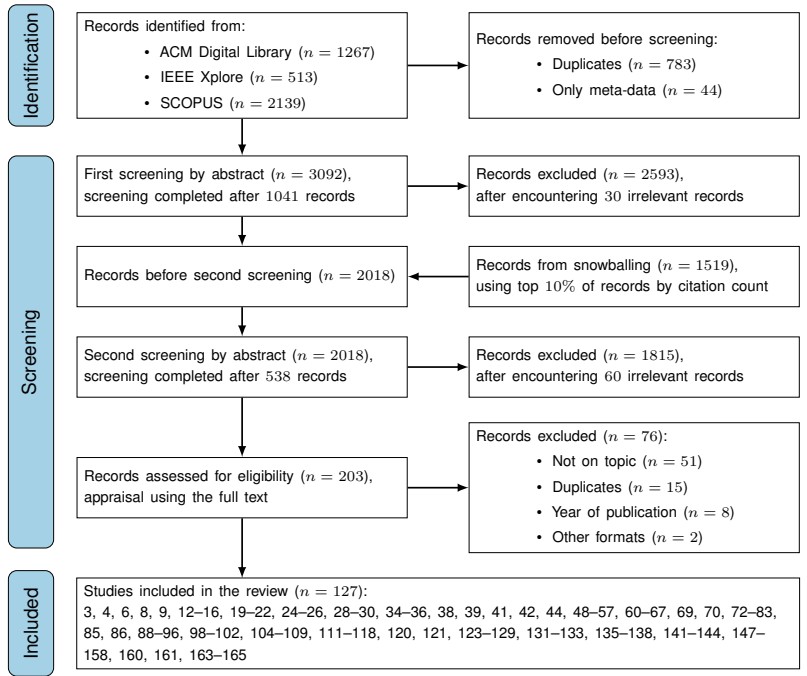

Figure 1: Identification of studies via databases and snowballing

## 3.3 Synthesis

To compile the results we carry out a standard thematic content analysis following the approach presented in [40]. First, we explore the data extracted from the set of publications relevant to each question to find the commonalities, which serves as the grounds for creating the initial set of codes. We evaluate the documents against these codes and keep track of any other considerations. If such considerations appear in multiple documents, we create new codes for them. Afterward, we re-evaluate all documents against the new code. As the coding exercise is carried out by one author, they do a third pass over all documents to double-check for potential errors. Finally, where relevant, we cluster the codes into larger themes. In this analysis we only look at the explicit statements provided by the authors, we do not attempt to infer their understanding of the problem. Thus, the numbers provided in Section 4 should be understood as describing how algorithmic recourse is *discussed* in the literature. For brevity, we focus our discussion on the main themes, but we still highlight specific publications if we observe that the authors introduce novel, highly relevant considerations that do not fit into other themes. Finally, even though we also evaluated the technical aspects of the proposed solutions – requirements for methods and datasets used in evaluations – they are not covered in this review. Instead, we point the interested readers to [48, 70, 143].

## 4 Results

The following nine sections introduce the results of the thematic analysis. For each question, we explain why it is relevant to the analysis and examine the main themes. We also highlight highly important but underexplored themes. We start with the general points such as contributions and definitions in Sections 4.1 to 4.3. Then, in Sections 4.4 to 4.7 we investigate the societal components of AR research. Finally, in Sections 4.8 and 4.9 we look at the aspects relevant to practitioners.

### 4.1 What types of contributions do the authors choose to make to the AR research?

We start by looking at the main goals of the collected publications to validate our assumption that AR literature is primarily concerned with technical solutions. We annotate each entry with at most two codes based on the form of contributions. By far the largest group is *propose methods*, which applies to 88 (69.3%) out of the 127 publications. These are primarily generators for individual CEs, but we also find 18 (14.2%) documents that propose other methods. Next, 20 (15.7%) publications *develop theoretical frameworks*, for instance by grounding AR in user studies or providing critical perspectives on the problem. Further, 15 (11.8%) focus on *empirical or theoretical analyses* of the properties of AR and another 15 publications *apply* it in a variety of domains. We did not identify any applications evaluated with humans in the loop. Then, 5 (3.9%) publications *benchmark* existing methods, while 3 (2.4%) *review* them. We make our annotations available in technical Appendix B.

### 4.2 What are the criteria covered in the authors' definitions of AR?

We also evaluate what is understood as the problem to be addressed by AR mechanisms. In particular, what are the criteria to satisfy authors' definitions of recourse. A similar question was posed by [70] who combined six definitions into *"recourse can be achieved by an affected individual if they can understand and accordingly act to alleviate an unfavorable situation, thus exercising temporally-extended agency"*, but this approach was far from systematic. Instead, we are interested in the underlying concepts. 74 (58.3%) publications explicitly define AR, 16 (12.6%) mention it but do not include a definition, while 37 (29.1%) do not mention AR, even though they align with its (overall) goals. The most common theme is *overturning undesirable decisions*, present in 47 definitions (63.5% of all definitions), but specifically *overturning algorithmic decisions* is mentioned only 43 (58.1%) times. It is generally understood that *AR is provided to affected individuals* (44, or 59.5%) but 4 (5.4%) definitions *consider stakeholders* more broadly. *Actionability* as a requirement for recourse is noted in only 39 (52.7%) definitions. Then, 20 (27.0%) publications specifically mention counterfactual explanations as means to AR, while 26 (35.1%) include various other technical considerations in the definitions, such as "changes to actionable input variables" or "desired classes".

We also point to several themes that are, interestingly, underrepresented. Only 18 (24.3%) documents mention *explanation, justification, or understanding of a decision* as the pre-requisite for AR. Next, 10 (13.5%) highlight *future-orientation or other temporal aspects* of the provided recommendations. Although *"consequential settings"*, typically bank lending, are given as examples in nine (12.2%) definitions, they are never explicitly mentioned as the scenarios where recourse ought to be provided, which may be akin to the "enjoyment of recourse" as defined by [142] where people are aware that there exists a way to reverse undesirable decisions.[1] 8 publications (10.8%) promote *AR as an ability*. Finally, only 2 (2.7%) publications require that recourse accounts for the *preferences* of its recipients.

### 4.3 What are the criteria covered in the authors' definitions of actionability?

As we observe, "actionability" is a concept that underpins AR but we discover that, in general, its understanding is limited. 91 (71.6%) publications attempt to define what it means (for a CE) to be actionable. Most commonly, in 48 (52.7%) out of 91 definitions, it is understood as *acting only on directly-mutable features*, 6 (6.6%) distinguish that *features may be indirectly-mutable* but still not actionable, while 22 (24.2%) also highlight that *feature values may need to be constrained*. Next, 19 (20.9%) definitions rely on a tautology that actionability means *people can take actions*, 11 (12.1%) emphasize that these *actions must be successful or lead to change*, and 3 (3.3%) further require that they are *aligned with people's real-world objectives*. Only 14 (15.4%) definitions put users

---

[1]Financial domain dominates the evaluations as well, with 90 of 116 evaluations on non-synthetic data making use of at least one finance-related dataset, most commonly `German Credit Data` [59] with 51 uses.

at the center stage, indicating that actionability *depends on the user or their preferences*, while 2
(2.2%) highlight the *importance of the context* [144, 156], for instance, that the ability to act on a
recommendation may change over time. Importantly, ethical considerations are never mentioned as
the pre-requisite for actionability, but we find some broader discussions about this [e.g., 142].

## 4.4 What is the role of end users? What other stakeholders are envisioned in the AR process?

Given that AR is to be implemented in socio-technical systems that include a variety of actors, we
are interested in the types of stakeholders acknowledged in the literature. A total of 105 publications
provide explicit consideration of this type. In general, end users subject to algorithmic decisions
are envisioned to be the recipients of AR, but this is not always the case: it may also be provided to
experts [e.g., 21, 22, 76] or organizations [e.g., 65, 72, 147], which highlights that in some cases AR
may be carried out on behalf of the affected individuals. In any case, 47 (44.8%) publications in the
subset agree that end users should inform actionability, but it is rarely clear *how* these preferences
should be specified. User-friendly (interactive) interfaces are a consideration in only 14 (13.3%)
documents. A total of 29 (27.6%) publications envision domain experts as someone who inform
the recourse process. They are either expected to inform actionability in the AR system or provide
other forms of knowledge, typically in the form of a causal structure. Besides the experts, authors
of 35 (33.3%) papers have discussed a variety of stakeholders. Most commonly system owners
[e.g., 20, 34, 38, 89], but also auditors [e.g., 138, 158], data scientists [e.g., 28, 82], developers [e.g.,
22, 131], practitioners [e.g., 100, 156], regulators [e.g., 28, 120], or even potential attackers [102].

## 4.5 What types of real-world considerations motivate existing research?

With the multitude of challenges that stand ahead of real-world AR, we are interested in the considera-
tions that motivate existing work. The main theme we find is *ensuring proper individual actionability*,
which is addressed in 46 (37.4%) of 123 publications relevant to this question. This is typically
achieved with the encoding of user preferences as constraints, but other means include providing
diverse CEs. In fact, *tackling specific desiderata for AR* (beyond actionability) is the second largest
area of research with 28 (22.8%) publications. Various *other technical challenges* are considered
in 24 (19.5%) documents, for example, integrating background knowledge [e.g., 16, 62, 64, 98], or
incorporating feature importance [e.g., 4, 6, 96, 116]. We also find 19 (15.4%) publications that
discuss the problem of *communicating recourse to the end users*. 16 (13.0%) focus on the *dynamics
of real-world systems*, typically addressing the robustness of AR [e.g., 75, 91, 93, 137], while 14
(11.4%) look at recourse in *multi-agent systems*. This also relates to *performance considerations*
emphasized in 15 (12.2%) of documents. *Causality* drives research in 14 (11.4%) cases. We also
find several themes that are under-emphasized: only 9 (7.3%) publications are directly *motivated by
research in psychology*, while *ethics of AR* are emphasized in only 7 (5.7%) documents.

## 4.6 What types of real-world considerations are seen as challenges for future work?

While the previous section looked at the considerations that drive existing research, in this section we
distill the recommendations for *future* research going beyond the improvement of own work, which
are provided in 74 documents. *Causality* is highlighted as a challenge in 22 (29.7%) of them, while
*other technical considerations* are given in 20 (27.0%) cases. These range from robustness [e.g.,
51, 117, 137], support for categorical features [e.g., 36, 157], or distinguishing between valid CEs and
adversarial examples [101]. Next, 19 (25.6%) documents highlight the importance of *ensuring proper
individual actionability*, which also relates to *communicating recourse to the end users* (9, or 12.2%)
and *supporting realistic cost functions* (8, or 10.8%). *Ethics of AR* are highlighted in 11 (14.9%)
publications, for example, that AR research may detract from other obligations of model owners
[77, 133]. The same number of publications emphasize the need to (1) *ground research in user studies*,
and (2) accommodate for the *dynamics of real-world systems*. *Privacy or security* is highlighted in 10
(13.5%) documents, while the *abuse of recourse*, such as strategic behaviors, surfaces in 7 (9.4%)
papers. Other challenges include improving *performance* (8, or 10.8%), considering *multi-agent
systems* (4, or 5.4%), and developing *legal frameworks* (4, or 5.4%) for recourse. We also highlight
several challenges particularly relevant to our work: (the usefulness of) recourse is perceived as
difficult to evaluate in practice [41, 60, 115], it must account for individual, contextual, societal, and
even cultural factors [123], which further means that engagement with recourse mechanisms and the
likelihood of its implementation are context-dependent [e.g., 6, 42, 128].

## 4.7 What types of (emergent) group-level dynamics are addressed in the existing research?

Real-world systems entail the implementation of recourse by more than one agent, which may introduce group-level dynamics. Nonetheless, out of 119 documents relevant to this question, 93 (78.2%) seem to understand recourse as a purely individual phenomenon. Among the remaining 26 documents we find considerations for several different group-level effects. Various perspectives on the problem of fair AR, covering both individual and group formulations are addressed by [12, 36, 52, 120, 121, 131, 149, 154]. Next, [9] shows that the implementation of AR on a large scale may lead to domain and model shifts, which introduce unexpected costs for the stakeholders.[2] In [42] the authors focus on another negative consequence of AR at scale, showing that it may reinforce social segregation. The impact of the "right to be forgotten", where data deletion requests trigger model retraining that may invalidate existing recourses is addressed in [75]. Then, [94] develop a game-theoretic framework for AR in multi-agent settings, attempting to optimize for "social welfare" rather than the profits of individual agents. We find two further similar perspectives on recourse: [38] proposes auditing and subsidies to minimize the risks of strategic behaviors in a multi-agent setting, while [136] attempts to incentivize actual improvement for a population of agents. Finally, [65] provides a framework that generates transparent and consistent recourses for a sub-population. We also note two other lines of research that account for the remaining documents with group-level considerations. First, in a causal setting [e.g., 68, 73] subpopulations are necessary to estimate the interventional effects on individuals. Second, several works highlight the importance of global insights into the data [22, 41, 44, 78, 108, 112, 152], such as recourse summaries [78, 112].

## 4.8 What are the approaches to the realistic evaluation of proposed methods?

We now explore the different forms of "real-world" evaluations, going beyond quantitative experiments, which are present in 51 publications. Most commonly, in 28 (54.9%) of those, the authors make use of *case studies* presenting the methods in an end-to-end manner. Among those, the application of recourse in the `Hired.com` marketplace goes furthest in simulating real-world conditions for AR [89], but the recommendations are still not evaluated with humans in the loop. Further, 9 (17.6%) documents include other forms of *short walk-through examples*. We also identify 14 (27.5%) papers that evaluate the methods with *user experiments*, 10 of which involve non-expert users and 4 involve expert users. While we do not observe any interviews with non-expert users, we find 1 (2.0%) publication where *experts are interviewed* [22]. *Other involvement of non-experts* applies to [116], where they inform the development of methods. *Other involvement of experts* is featured in two documents where they evaluated the outputs of methods [25, 132]. Altogether, end users were involved in 17 publications, which is only 13.3% of all publications covered in our study, even more striking than the 21% of CE methods evaluated with user studies as reported in [71].

## 4.9 What are the open source and documentation practices in AR research?

Finally, we note that the lack of availability of well-documented open-source code may be an important obstacle to the application of AR in real-world systems. For all 116 publications that involve some form of computational experiments, we verify whether the source code is publicly available. If the authors do not explicitly link to their code in the paper, we attempt to find it independently. Ultimately, we collect open-source implementations for 64 (55.2%) publications. Then, for each of them, we evaluate the quality of documentation. The *instructions on the general usage* (such as installation and workflow) are provided with 27 (41.5%) repositories, while *instructions on the reproduction of results* in 23 (35.4%). In 19 (29.2%) cases we find *walk-through tutorials*, typically in the form of Jupyter Notebooks, although we note that they differ in quality. For instance, 5 repositories include code-only notebooks with no further textual explanation that could guide the practitioner. Implementations for 4 papers include more "professionalized" *documentation* [9, 86, 100, 156]. The latter sets a golden standard as it further includes a tutorial video and a live demo. We do not find *any* additional materials for practitioners for 13 (20.0%) of the available implementations.

---

[2]Such "endogenous dynamics" were postulated earlier in the first version of [113] dated December 22[nd] 2020, but this discussion has been completely removed from the subsequent versions of the pre-print.

# 5 Discussion

Regardless of whether AR can be normatively expected or not [77], many systems can genuinely benefit from recourse mechanisms, especially when the interests of the system owner and the end users are aligned [72], such as in the healthcare system to improve the well-being of patients [76, 96, 155], or on the online platforms that attempt to improve the experience of their users [89, 134]. Nonetheless, the values and norms underlying recourse – trust, agency, fairness, safety, and so on – are emergent properties of systems where recourse mechanisms would be introduced. Such norms can only be understood and evaluated when accounting for the technical, social, and institutional components of the system [32], but the latter two remain largely unexplored in the recourse literature.

Recourse is not inherently safe or unsafe, *but* its (incorrect) implementation may lead to the emergence of unsafe dynamics, such as the unexpected costs to stakeholders as discussed by [9] or the reinforcement of social segregation addressed in [42]. While it may be too challenging to provide accurate system-level evaluations at this stage of research, authors can still expand the boundaries of their analyses to account for global effects or look at the position of recourse mechanisms in the broader context of a complete socio-technical AI system [33]. As AR is a "reality-centric AI" problem [140] by its nature, working towards its integration into existing systems will require a design-oriented approach, potentially with *specific* systems in mind. The "Abstraction Traps" discussed by [119] in the context of research on fair machine learning apply here: that technical solutions designed for one social context cannot be directly repurposed for another application, that values to which they are expected to adhere to cannot be captured with mathematical formulas, that their insertion into an existing process will impact its behavior, or that the best solutions may not necessarily be technical.

It is perhaps most telling that only 12% of surveyed publications attempt to apply recourse in realistic settings. We will discuss two of these settings to highlight the stark differences in system properties. Most of the applications included in our review focus on the provision of actionable individual recommendations to students [3, 4, 24, 109, 126, 135, 160]. In this relatively low-stakes domain almost any recourse will be actionable in that following a personalized set of learning activities does not require any resources other than time. Even then, the system involves multiple actors – students, teachers, parents – whose interactions will impact the process, for example, because students may fail to benefit from certain learning activities without additional support. Conversely, we find several publications where authors attempt to provide recourse in the high-stakes medical domain [76, 96, 155]. Here, recommendations must be tailored to the preferences, resources, or lifestyles of patients in order to have a chance of being actionable. Moreover, certain aspects of their implementation fully rely on other actors, such as a clinician prescribing the medications. Finally, it may happen that recourse does not exist at all when the outcomes of a patient cannot be improved.

## 5.1 Recommendations for future research

We distill our findings into five key recommendations. First, in Sections 4.2, 4.3 we observed that *operational* definitions for recourse are still unavailable. Second, Sections 4.4 and 4.8 underlined little consideration for people involved in recourse processes. Third, Sections 4.5, 4.6 highlighted the overwhelmingly technical approaches to recourse. Fourth, Section 4.7 stressed the lack of group-level analyses. Fifth, from Sections 4.8, and 4.9 we learned about the missing consideration of practitioners.

**1. Broadening the scope of research.** AR is generally seen as a service for affected individuals, but this formalization may be unnecessarily limiting. In fact, in many systems, these individuals may be unable to directly act on recommendations [see also 142]. Instead, we propose to operationalize the aim of AR as the provision of recommendations *aligned with the preferences* of *non-expert users* in an attempt *to help them improve outcomes* in an *ADM setting*, which emphasizes that providing *easy to understand* and *individually actionable* recommendations remains the key research problem.

**2. Engaging end users, affected individuals, and communities.** AR solutions are rarely evaluated with humans. Instead, they attempt to satisfy a variety of desiderata formulated by authors and assessed in an automated manner. Sparsity, proximity, or mutability of features are far from perfect proxies for individual actionability. For AR to be truly useful, it must be able to satisfy the preferences of its end users. Research is also necessary to learn about the needs of the affected individuals concerning recourse, and to validate its potential contributions and inherent limitations. Authors may also benefit from the rich literature on human-computer interaction [e.g., 11, 23] or psychology.

**3. Accepting a socio-technical perspective.** A pervasive assumption in the literature is that all challenges of AR require purely technical solutions. For instance, many authors emphasize the importance of causal modeling to guarantee recourse, but the models that aim to be explained are themselves *not* causal. Similarly, to improve the performance of CE generators many authors turn to deep generative models [35, 42, 61, 67, 81, 90, 99]. Not only do they explain the data rather than the model [10], but more importantly they shift the problem from improving the trust in non-interpretable models, to attempting to trust non-interpretable explainers. Although a socio-technical perspective on AR brings its own challenges, such as accounting for the roles of stakeholders involved in the provision of recourse, it creates important opportunities. For example, developing "recourse contracts" [34, 39] or designing feedback processes to account for imperfect robustness.

**4. Accounting for emergent effects.** Decision-making systems involve multiple individuals who may be interested in receiving recourse and may have competing interests. Research on AR should, from the onset, explore group-level effects such as external costs or fairness. While this may require expanding the boundaries of analysis, it is necessary to anticipate the emergent outcomes of recourse. These may even occur due to the multi-system dynamics of AR: recommendations implemented by an individual to improve their outcomes in one system will affect them in other contexts [see also 13].

**5. Attending to other operational aspects.** Finally, the artifacts of AR research should be practitioner-friendly. On the one hand, this requires being explicit about the position of the proposed methods in a broader system, for example, in the form of end-to-end case studies that allow practitioners to better understand the benefits of the proposed solutions. On the other hand, this suggests that authors should attempt to move away from merely providing scripts for experiments, and focus on developing well-documented frameworks that can be adapted to different ADM systems.

## 5.2 Limitations of our work

Our review is not without shortcomings. Most importantly, for each paper the extraction and coding of data was performed by a single author, which means that the quantitative results may be imperfect. We account for this by focusing the analysis on the *overarching themes* represented in existing publications, thus, even if another researcher would have carried out the coding in a somewhat different manner, they should arrive at similar results and our analysis remains valid. Additionally, as our review ultimately looks at the authors' perception of recourse, we do not want to misconstrue their views. Thus, we do not infer any considerations unless they are provided explicitly. Our reading may be more strict than intended by the authors and the numbers reported in our results may be underestimated. At the same time, we believe that if certain considerations are deemed important by the researchers, they would choose to be explicit about them. Finally, although we followed a systematic process, we cannot claim that we collected AR literature in an exhaustive manner due to the specificities of computer science publishing. Thus, we acknowledge that there may exist some insightful publications addressing recourse that have not been covered in this literature review.

## 6 Conclusions

Algorithmic recourse concerns the provision of recommendations aligned with the preferences of non-expert users of algorithmic decision-making systems to help them achieve more desirable outcomes in the future. Existing research on the topic is predominantly theoretical, even though recourse, in expectation, is a real-world problem with strong practical implications. To that end, we conducted a systematized literature review of 127 publications that focus on algorithmic recourse, and more generally on actionable counterfactual explanations. We evaluated the practical considerations provided by the authors. Our findings indicate that, indeed, AR tends to be perceived as a (predominantly) technical problem. Although we think highly of fundamental research, we note that for algorithmic recourse to leave computer science labs, it must be more strongly grounded and validated in the real world, and consider the requirements for systems that include not only technical but also social and institutional components. To help bridge this gap, we synthesize a list of five recommendations for other authors that aim to reinforce recourse as a practical problem. We believe that AR should not be seen as only a simple ad-hoc solution to improve the acceptance of black-box models in consequential domains, but rather as a full-fledged socio-technical mechanism that can benefit many systems and improve the agency of affected individuals and decision-makers across a variety of settings.

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

## A    Extended discussion of the search process

While our discussion of the search process in Section 3.1 in the main body of the document is complete, we also provide an extended version of this discussion to allow for full reproducibility.

We make use of 3 search engines to collect the initial set of studies: ACM Digital Library, IEEE Xplore, and SCOPUS. Given the blurry distinction between AR and CEs, we consider the papers discussing either problem. In a small scoping review, we identify several keywords common to publications on recourse, as well as several equivalent terms to build the query shown below.

```
(''Machine Learning'' OR ''Artificial Intelligence''
OR ''Algorithmic Decision*'' OR ''Consequential Decision*''
OR Classif* OR Predict* OR ''Explainable AI'' OR AI OR XAI)
AND (((Counterfactual OR Contrastive OR Actionable) AND Explanation*)
OR ((Algorithmic OR Individual* OR Actionable) AND Recourse)
OR Counterfactual?)
```

We modify this query to account for the semantic differences between the search engines.

For ACM Digital Library:

```
Title:(( "Machine Learning" OR "Artificial Intelligence"
OR "Algorithmic Decision*" OR "Consequential Decision*"
OR classif* OR predict* OR "Explainable AI" OR ai OR xai )
AND ( ( ( counterfactual OR contrastive OR actionable )
AND explanation* ) OR ( ( algorithmic OR individual* OR actionable )
AND recourse ) OR counterfactual? ))
OR Abstract:(( "Machine Learning" OR "Artificial Intelligence"
OR "Algorithmic Decision*" OR "Consequential Decision*"
OR classif* OR predict* OR "Explainable AI" OR ai OR xai )
AND ( ( ( counterfactual OR contrastive OR actionable )
AND explanation* ) OR ( ( algorithmic OR individual* OR actionable )
AND recourse ) OR counterfactual? ))
OR Keyword:(( "Machine Learning" OR "Artificial Intelligence"
OR "Algorithmic Decision*" OR "Consequential Decision*"
OR classif* OR predict* OR "Explainable AI" OR ai OR xai )
AND ( ( ( counterfactual OR contrastive OR actionable )
AND explanation* ) OR ( ( algorithmic OR individual* OR actionable )
AND recourse ) OR counterfactual? ))
```

For IEEE Xplore:

```
((("All Metadata":"Machine Learning"
OR "All Metadata":"Artificial Intelligence"
OR "All Metadata":"Algorithmic Decision*"
OR "All Metadata":"Consequential Decision*"
OR "All Metadata":classif* OR "All Metadata":predict*
OR "All Metadata":"Explainable AI" OR "All Metadata":ai
OR "All Metadata":xai )
AND ((("All Metadata":counterfactual OR "All Metadata":contrastive
OR "All Metadata":actionable ) AND "All Metadata":explanation* )
OR ( ("All Metadata":algorithmic OR "All Metadata":individual*
OR "All Metadata":actionable )
AND "All Metadata":recourse )
OR "All Metadata":counterfactual? )))
```

For SCOPUS:

```
TITLE-ABS-KEY ( ( "Machine Learning" OR "Artificial Intelligence"
OR "Algorithmic Decision*" OR "Consequential Decision*"
OR classif* OR predict* OR "Explainable AI" OR ai OR xai )
AND ( ( ( counterfactual OR contrastive OR actionable ) AND explanation* )
OR ( ( algorithmic OR individual* OR actionable ) AND recourse )
OR counterfactual? ) )
```

The search is carried out on January 12th 2024 in titles, abstracts, and keywords, with 1267 results from ACM Digital Library (The ACM Guide to Computing Literature), 513 results from IEEE Xplore, and 2139 results from SCOPUS. This leads to a total of 3919 results, which are imported to the Zotero reference management software for de-duplication. After removing the duplicates, we are left with 3136 results, 44 of which are the meta-data of conference proceedings that we also remove.

To facilitate the screening process, we employ the open-source ASReview tool, which makes use of an active learning approach to re-order the set of publications, such that the most relevant ones are always "at the top of the stack" [139]. We run ASReview on the default settings, i.e.:

```
Feature extraction technique: TF-IDF
Classifier: Naive Bayes
Query strategy: Maximum
Balance strategy: Dynamic resampling (Double)
```

The researchers behind the tool suggest employing a stopping rule measured in the number of consecutive irrelevant records, which we set to 30, or 1% of the entire dataset. We accept all papers that focus on algorithmic recourse and counterfactual explanations, completing the screening after evaluating 1040 abstracts (33.67% of the dataset), leading to 504 (16.30%) records among which we identify further 4 duplicates to remove. This results in the reported number of 499 relevant records.

We observe that some important publications may be missing from our results. For instance, [151] was published in the Harvard Journal of Law & Technology that is not indexed by computer science search engines. Thus, we decide to augment the set of records by applying snowballing, which has been shown as a good alternative to databases in systematic reviews in software engineering [162].

We decide to make use of citation counts as a proxy for impact. Due to the lack of a suitable tool that would provide unbiased citation counts for *all* papers in our dataset, we collect them from Google Scholar. Unfortunately, citation counts on Google Scholar tend to be inflated, but as we make use of snowballing purely to enrich the dataset, these does not impact the validity of our study. We manually collect Google Scholar citation counts for all 499 results from the first screening on January 27th and 28th, order them descendingly, and collect references for the top 50 (10%) "most impactful" publications. Snowballing results in a total of 1519 new records. Indeed, we observe that [151] (mentioned above) is referenced by 39 of the 50 publications used for snowballing.

While this strategy introduces several pre-prints into our result set [52, 61, 91, 113, 143, 150], we decide not to exclude them. Our review remains primarily concerned with peer-reviewed work. Here, we also note that [114], which we collected as a pre-print has been published between the search and appraisal. As such we decided to evaluate its published version and refer to it in this paper.

After adding the snowballed references into our dataset, we are left with 2018 records for the second screening with ASReview, again on the default settings. This time, we look for publications that specifically refer to the problem of AR, "actionable" CEs, or modifying outcomes of automated decision-making systems. We employ a stricter stopping rule to minimize the risk of false negatives, completing the screening after 60 consecutive irrelevant records. We evaluate 538 results (26.71% of the dataset), with 203 (10.06%) relevant results that are considered for full-text appraisal. This concludes the extended discussion of the search process.

 # B   Evaluation of contributions

Table 1: Evaluation of the collected publications on the types of contributions, 2017-2021.

| Year | Reference | Propose methods | Theoretical frameworks | Analyses | Apply | Benchmark | Review |
|------|-----------|-----------------|------------------------|----------|-------|-----------|--------|
| 2017 | [151] | ✓ | ✓ | | | | |
| 2019 | [52] | ✓ | | | | | |
| | [61] | ✓ | | | | | |
| | [81] | ✓ | | | | | |
| | [85] | ✓ | | | | | |
| | [138] | ✓ | | | | | |
| 2020 | [35] | ✓ | | | | | |
| | [86] | ✓ | | | | | |
| | [136] | ✓ | | | | | |
| | [20] | ✓ | | | | | |
| | [26] | ✓ | | | | | |
| | [44] | ✓ | | | | | |
| | [67] | ✓ | | | | | |
| | [66] | ✓ | | | | | |
| | [99] | ✓ | | | | | |
| | [104] | ✓ | | | | | |
| | [107] | ✓ | | | | | |
| | [120] | ✓ | | | | | |
| | [112] | ✓ | | | | | |
| | [13] | | ✓ | | | | |
| | [142] | | ✓ | | | | |
| 2021 | [69] | ✓ | ✓ | | | | |
| | [137] | ✓ | | ✓ | | | |
| | [41] | ✓ | | | | | |
| | [49] | ✓ | | | | | |
| | [53] | ✓ | | | | | |
| | [73] | ✓ | | | | | |
| | [150] | ✓ | | | | | |
| | [105] | ✓ | | | | | |
| | [19] | ✓ | | | | | |
| | [22] | ✓ | | | | | |
| | [63] | ✓ | | | | | |
| | [64] | ✓ | | | | | |
| | [88] | ✓ | | | | | |
| | [98] | ✓ | | | | | |
| | [115] | ✓ | | | | | |
| | [117] | ✓ | | | | | |
| | [153] | ✓ | | | | | |
| | [161] | ✓ | | | | | |
| | [121] | ✓ | | | | | |
| | [55] | | ✓ | | | | |
| | [12] | | | ✓ | | | |
| | [113] | | | ✓ | | | |
| | [125] | | | ✓ | | | |
| | [4] | | | | ✓ | | |
| | [82] | | | | ✓ | | |
| | [89] | | | | ✓ | | |
| | [96] | | | | ✓ | | |
| | [135] | | | | ✓ | | |
| | [152] | | | | ✓ | | |
| | [160] | | | | ✓ | | |
| | [100] | | | | | ✓ | |

Table 2: Evaluation of the collected publications on the types of contributions, 2022.

| Year | Reference | Propose methods | Theoretical frameworks | Analyses | Apply | Benchmark | Review |
|------|-----------|-----------------|------------------------|----------|-------|-----------|--------|
| 2022 | [39] | ✓ | | ✓ | | | |
| | [34] | ✓ | | ✓ | | | |
| | [6] | ✓ | | | | | |
| | [25] | ✓ | | | | | |
| | [50] | ✓ | | | | | |
| | [62] | ✓ | | | | | |
| | [158] | ✓ | | | | | |
| | [83] | ✓ | | | | | |
| | [56] | ✓ | | | | | |
| | [79] | ✓ | | | | | |
| | [80] | ✓ | | | | | |
| | [90] | ✓ | | | | | |
| | [93] | ✓ | | | | | |
| | [106] | ✓ | | | | | |
| | [111] | ✓ | | | | | |
| | [132] | ✓ | | | | | |
| | [131] | ✓ | | | | | |
| | [144] | ✓ | | | | | |
| | [65] | ✓ | | | | | |
| | [101] | | ✓ | ✓ | | | |
| | [24] | | ✓ | | ✓ | | |
| | [70] | | ✓ | | | | ✓ |
| | [15] | | ✓ | | | | |
| | [16] | | ✓ | | | | |
| | [94] | | ✓ | | | | |
| | [118] | | ✓ | | | | |
| | [133] | | ✓ | | | | |
| | [157] | | ✓ | | | | |
| | [128] | | ✓ | | | | |
| | [149] | | | ✓ | | | |
| | [28] | | | | ✓ | | |
| | [109] | | | | ✓ | | |
| | [126] | | | | ✓ | | |
| | [48] | | | | | ✓ | ✓ |
| | [143] | | | | | ✓ | ✓ |

Table 3: Evaluation of the collected publications on the types of contributions, 2023-2024.

| Year | Reference | Propose methods | Theoretical frameworks | Analyses | Apply | Benchmark | Review |
|------|-----------|:---------------:|:----------------------:|:--------:|:-----:|:---------:|:------:|
| 2023 | [36] | ✓ | ✓ | | | | |
| | [29] | ✓ | ✓ | | | | |
| | [116] | ✓ | ✓ | | | | |
| | [9] | ✓ | | ✓ | | | |
| | [42] | ✓ | | ✓ | | | |
| | [75] | ✓ | | ✓ | | | |
| | [147] | ✓ | | ✓ | | | |
| | [156] | ✓ | | | ✓ | | |
| | [155] | ✓ | | | ✓ | | |
| | [54] | ✓ | | | | | |
| | [123] | ✓ | | | | | |
| | [14] | ✓ | | | | | |
| | [72] | ✓ | | | | | |
| | [30] | ✓ | | | | | |
| | [51] | ✓ | | | | | |
| | [91] | ✓ | | | | | |
| | [92] | ✓ | | | | | |
| | [95] | ✓ | | | | | |
| | [108] | ✓ | | | | | |
| | [127] | ✓ | | | | | |
| | [129] | ✓ | | | | | |
| | [141] | ✓ | | | | | |
| | [154] | ✓ | | | | | |
| | [163] | ✓ | | | | | |
| | [164] | ✓ | | | | | |
| | [165] | ✓ | | | | | |
| | [78] | ✓ | | | | | |
| | [148] | ✓ | | | | | |
| | [74] | ✓ | | | | | |
| | [77] | | ✓ | | | | |
| | [124] | | ✓ | | | | |
| | [38] | | | ✓ | | | |
| | [57] | | | ✓ | | | |
| | [102] | | | ✓ | | | |
| | [3] | | | | ✓ | | |
| | [76] | | | | ✓ | | |
| | [8] | | | | | ✓ | |
| | [60] | | | | | ✓ | |
| 2024 | [21] | ✓ | | | | | |
| | [114] | ✓ | | | | | |

