# OpenReview forum: "Grounding and Validation of Algorithmic Recourse in Real-World Contexts: A Systematized Literature Review"
_NeurIPS.cc/2024/Conference — Submitted to NeurIPS 2024_

### Official Review · Reviewer_Wg7R · 2024-07-05

**Soundness:** 2
**Presentation:** 3
**Contribution:** 2
**Rating:** 5
**Confidence:** 4

**Summary:**

This paper provides a review of previous works that study "algorithmic recourse", i.e. conceptual and practical approaches for giving people actionable recommendations to change how they are impacted by algorithmic systems. This literature is deeply connected with counterfactual explanations and understanding models through small changes to test data, answering questions such as "how would the model M produce a different output if changed attribute x about myself". The authors review 127 archival publications and answer 9 questions about how these works frame and study algorithmic recourse.

**Strengths:**

In terms of originality, quality, and clarity:
- While the primary novel contributions of this draft are to highlight themes in previous work, the overall level of novelty is reasonable. Some concerns here, see below.
- Quality: the "Systematized review" methods are described such that they are replicable and seem justified. I don't expect readers to have major issues with inclusion criteria of papers, or any of the analyses presented.
- Clarity: Writing is clear throughout.

In terms of significance, the paper could have impact on future work studying algorithmic recourse, and might motivate NeurIPS community members (including those in companies or working with governments) to support recourse methods. This would be a large positive impact.

 This kind of review can certainly be useful to researchers trying to incorporate ideas or findings from recourse-related research. The calls to engage with HCI and systems-level thinking are reasonable (though, some of the broader discussion/motivation in the paper is more convincing on this front than any of the empirical results from the 127 recourse-related papers). If a version of this paper were able to unify definitions in the recourse space, this could be powerful (though further expansion of Section 2.2 might be necessary: the paper does note that reference [70] is highly similar -- the current draft was a bit vague in comparing these and clarifying the added contribution here.).

A few other notes: There are 9 overall sub-research questions answered. Overall, these results seem likely to be useful to researchers entering the algorithmic recourse field (though, see below, some of these felt very general and not domain-specific in the current draft). The paper does fit into the "Social and economic aspects of machine learning" category listed in the CFP this year.

**Weaknesses:**

Overall, I do think the current draft may not achieve the full impact that a future revision could provide.

The current discussion section feels like it largely echoes other calls in the community to apply systems thinking to ethical/responsible/pro-social AI/ML initiatives, and while each of the suggestions has some connection to one of the analyses, the current draft is not totally clear about the extent to which these recommendations stem from the findings vs. are motivated by first principles. The paper is overall very critical of the AR field, i.e. "Why hasn't this field engagement with any real world deployments". However, it's also not entirely clear in the current draft how any of the general recommendations would be applied in specific AR domains.

One aspect of the paper that I think would have been most directly helpful to the NeurIPS audience in particular would be to take a stance on how recourse should be defined -- is the definition on line 62 "endorsed" by the paper? Is the "imagine a counterfactual input x*" an advisable approach to take for future work. Does this review support the definition, highlight core definitional or epistemic issues, etc.

Ultimately, given the intended goals of this draft, it seems success and impact (on top of the core empirical contribution provided by writing a systematized review) here are dependent on the ability for the provided recommendations to shape future research positively. While the five recommendations here could have a some positive impact, taking a stronger stance on the core definitions and framing of recourse "tasks" could have an even larger impact.

**Questions:**

The primary questions that I would pose to the authors would be:
- Is it possible to use the current "data" (i.e. selected papers) to provide more actionable domain-specific recommendations and/or pragmatic guidance about which contexts are likely to see real-world engagement with AR?
- What is needed to get organizations that develop and/or operate algorithmic systems to engage with recourse? Are there circumstances that the current literature treats as "futile"?

I think using the data that's already been collected and analyzed, focusing on domain-specificity of recommendations could go very far in strengthening the draft.

**Limitations:**

- No major concerns regarding unmentioned social impacts.
- Regarding the limitations of systematized literature review, the current draft discusses these reasonably.

---

> ### Author Rebuttal · Authors · 2024-08-06
>
> Thank you for this insightful review and feedback!
>
> As mentioned in the "global rebuttal" we cut some parts of Section 5 to fit within the nine-page limit. As we see it, we are able to address your points using the data that we have already collected and processed.
>
> ---
>
> > If a version of this paper were able to unify definitions in the recourse space, this could be powerful
>
> > is the definition on line 62 "endorsed" by the paper? (...)
>
> We quote the definition on line 62 as it comes from the work that introduced AR, but we believe that it misses out on some important aspects that have been brought up by later authors. Hence, on lines 360-361, we propose the following operational definition of recourse: *"the aim of AR [is] the provision of recommendations aligned with the preferences of non-expert users in an attempt to help them improve outcomes in an ADM setting"*. It accounts for the concepts identified in Section 4.2:
>
> 1. AR is a recommendation, i.e., a form of *"advice about what is best to do"* (from Cambridge Dictionary)
> 2. The recommendation should be (where applicable) aligned with the preferences of their recipients to facilitate actionability.
> 3. While most authors agree that AR is aimed at the end-users, we found arguments that it may be provided to other stakeholders (lines 224-226). Ultimately, the goal remains the same: it should be understandable to individuals who are not necessarily knowledgeable about algorithms.
> 4. The goal of AR is to improve outcomes (the most common theme among the definitions), but the ability to improve an outcome will depend on the context and the willingness of the individual to implement it, hence the word "attempt".
> 5. Finally, the ADM settings seem to be assumed given the nature of existing work.
>
> We do not define what "actionability" entails because this concept ultimately depends on factors such as the domain, the context, or the stakeholders (see our answer to Reviewer pd5W).
>
> ---
>
> > the paper does note that reference [70] is highly similar -- the current draft was a bit vague in comparing these and clarifying the added contribution here.
>
> Our draft is distinct from the earlier literature reviews in that we take a step back from evaluating technical *solutions*, instead focusing on the practical understanding of the AR *problem*. We explain the differences with [70] further in the response to Reviewer pd5W.
>
> ---
>
> > Is it possible to use the current "data" (i.e. selected papers) to provide more actionable domain-specific recommendations and/or pragmatic guidance about which contexts are likely to see real-world engagement with AR?
>
> Very good point! Yes, we can address this based on our data. In the current draft, we briefly touch upon the aspects that should be considered for specific applications in Section 5. More concretely, one of the goals of our recommendations is to encourage thinking about the meaning of AR for a specific domain, e.g.:
>
> * [Recommendations. 1 and 3] What (types of) stakeholders will interact with the recourse mechanisms?
> * [Rec. 2] What constraints need to be satisfied to ensure that recourse can fulfill its goals?
> * [Rec. 3] What procedures are already in place? What will be the added value of recourse?
> * [Rec. 4] What (unsafe) group-level dynamics should we expect when recourse is implemented into the system?
> * [Rec. 4] If we should expect unsafe group-level dynamics, how can we mitigate them?
> * [Rec. 5] Which existing AR solutions could be applied in this domain? How to tailor them to its specific requirements?
>
> Regarding the second part of the question, we can make some predictions from the data. On the one hand, we note that many researchers discuss recourse in banking: this goes from commonly used examples to the dominance of banking datasets in the evaluations, although the availability of high-quality datasets may influence these choices. On the other hand, we see a variety of domains in the applications and case studies: education, medicine, public administration, or employment.
>
> A common characteristic in the latter domains is that the model owners have shared interests with the end-users in achieving the best outcomes possible (see also lines 318-325), or other responsibilities towards them. This may stem, e.g., from legal acts such as Art. 22(3) of the European Union's GDPR which bestows the right to obtain human intervention in certain situations (see also [151]).
>
> Finally, there may even exist settings where recourse adds value to ADM systems using models that tend to be perceived as "explainable". For instance, governance typically employs "decision trees" that implement rules from legislation, but even these trees can grow to a point where they are hard to understand (e.g., tax systems).
>
> ---
>
> >What is needed to get organizations that develop and/or operate algorithmic systems to engage with recourse? Are there circumstances that the current literature treats as "futile"?
>
> This question is perhaps best answered by looking at the challenges for future work (Section 4.6). Several major categories — accounting for causality, robustness, or even ensuring actionability — relate to a problem that will be highly relevant for the engagement with recourse in real-world contexts: how can we ensure that the recommendations lead to meaningful improvement *and* guarantee a better outcome for the individual? This results in Rec. 3, which states that organizations may want to consider societal solutions as well.
>
> We have not identified any circumstances that would be considered completely "futile" in that for all challenges recognized in the literature, we have observed at least some suggestions on how they could be resolved, although not necessarily extensive follow-up work (yet). In our opinion, AR should not be treated as a cure-all solution for problems brought upon by non-interpretable models. This insight tends to be missing in the literature, even though it does not make AR research any less valuable.

---

> > ### Comment · Reviewer_Wg7R · 2024-08-08
> >
> > Thank you for these detailed answers to the questions. This rebuttal
> >
> > - answers above questions about defining / redefining recourse. I think this can easily be improved in a camera ready
> > - clarifies relation to prior work
> > - provides quite a few a concrete additional empirical insights that can be added to flesh out some additional aspects of the paper
> >
> > Overall, this addresses some of my concerns. I expect the main lingering concern will be the question of venue fit. As argued in the global rebuttal, I can see this kind of paper fitting into the "catch all" category. Furthermore, as noted in my original review I agree this is a strong argument for trying to publish this work in venues that are most likely to impact recourse implementations in practice. That said, I expect this may come down to subjective interpretation of the CFP here.

---

> ### Author Response · Authors · 2024-08-09
>
> Thank you for the reply and the increased score! We are very happy that our answers satisfied your concerns about the contents of the draft — your suggestions have been very helpful, we have no doubts that going further in the recommendations will strengthen the document.
>
> We also appreciate your repeated support for our choice of the venue.
>
> Should you have any further questions, please let us know.

---

### Official Review · Reviewer_yANZ · 2024-07-09

**Soundness:** 2
**Presentation:** 3
**Contribution:** 2
**Rating:** 3
**Confidence:** 5

**Summary:**

The paper provides a comprehensive review of the algorithmic recourse research literature, concentrating on understanding the recourse research "in the wild", by focusing on the practical application of these techniques in real-world scenarios. The authors then provide some suggestions to practitioners to push future research to better practical applications.

**Strengths:**

The paper is well-written and well-structured. Considerable effort has been put into this work to provide a comprehensive review of the area, highlighting the need for a more down-to-earth approach when considering recourse. The data collection and analysis are well-motivated and described sufficiently (Section 3 and Section 4). The recommendations in Section 5.1 are on point and all true, and they highlight issues that everyone in the community is aware of but that are largely ignored.

**Weaknesses:**

I feel NeurIPS is not the right venue for this kind of contribution, since this paper does not provide the level of technical novelty required by the conference. Being a review, I think it does not fit the requirement of "new and original research" given by the Call of Papers. I suggest the authors not be discouraged, since I think the contribution is still valuable for the community. Potential other venues I believe are more in line with the scope of this work could be the following (the order is random):
- IJCAI Survey Track (https://ijcai24.org/call-for-papers-survey-track/)
- ACM FAccT (https://facctconference.org/)
- AAAI/ACM AIES (https://www.aies-conference.com/2024/)
- ICML Position Papers Track (https://icml.cc/Conferences/2024/CallForPositionPapers)
- ACM Computing Surveys (https://dl.acm.org/journal/csur)
- TMLR (https://jmlr.org/tmlr/)

Lastly, I would like to point out some potential additional papers on algorithmic recourse which could complement some remarks made by the authors:
- Line 182 "We did not identify any applications evaluated with humans in the loop": there has been some development in providing human-in-the-loop algorithms to identify better recourse options:
  - [1] De Toni, Giovanni, et al. "Personalized Algorithmic Recourse with Preference Elicitation." Transactions on Machine Learning Research, https://openreview.net/forum?id=8sg2I9zXgO
- Recommendation 4, "Accounting for emergent effects": there has been some research regarding providing recourse to multiple individuals, where they are competing for a limited pool of resources, looking also at the fairness of these systems:
  - [2] Fonseca, João, et al. "Setting the right expectations: Algorithmic recourse over time." Proceedings of the 3rd ACM Conference on Equity and Access in Algorithms, Mechanisms, and Optimization. https://dl.acm.org/doi/pdf/10.1145/3617694.3623251
  - [3] Bell, Andrew, et al. "Fairness in Algorithmic Recourse Through the Lens of Substantive Equality of Opportunity." arXiv preprint arXiv:2401.16088, https://arxiv.org/pdf/2401.16088

I also point the authors to some new papers considering human-in-the-loop interfaces for recourse (Recommendation 1, Section 5.1):
- [4] Esfahani, Seyedehdelaram, et al. "Preference Elicitation in Interactive and User-centered Algorithmic Recourse: an Initial Exploration." Proceedings of the 32nd ACM Conference on User Modeling, Adaptation and Personalization. https://dl.acm.org/doi/pdf/10.1145/3627043.3659556
- [5] Koh, Seunghun, Byung Hyung Kim, and Sungho Jo. "Understanding the User Perception and Experience of Interactive Algorithmic Recourse Customization." ACM Transactions on Computer-Human Interaction. https://dl.acm.org/doi/pdf/10.1145/3674503

**Questions:**

I do not have any questions for the authors.

**Limitations:**

The authors have highlighted the limitations of their work in Section 5.2.

---

> ### Author Rebuttal · Authors · 2024-08-06
>
> Thank you for this feedback, we are very happy that you perceive our work as useful for the community!
>
> ---
>
> > I feel NeurIPS is not the right venue for this kind of contribution, since this paper does not provide the level of technical novelty required by the conference.
>
> We understand your reasoning and really appreciate the effort you took to propose other venues. As Reviewer pd5W brought up a similar concern, please see the "global rebuttal" for an explanation of why we decided to submit this publication to NeurIPS and still believe that it is a suitable venue.
>
> ---
>
> >  I would like to point out some potential additional papers on algorithmic recourse which could complement some remarks made by the authors
>
> Thank you for bringing these further publications to our attention. We checked them against the complete set of 3092 records that have been collected for the review. None of the papers occur in our database, so we are glad to say that their omission is not an error in the screening process. We will not include them in Section 4 (Results) to keep our analysis consistent, but we will make sure to highlight them in the other sections of the final draft!

---

> > ### Comment · Reviewer_yANZ · 2024-08-12
> >
> > Thank you for the clarifications provided in the rebuttal. After reading the rebuttal and the other reviewers, I am still unconvinced that NeurIPS is the right venue for this contribution, so I will not change my evaluation.
> >
> > My main concern is still the lack of novel methodological contributions required by the Call for Papers. I praise the work done by the authors, but the points raised by the paper (although all true) are far from giving a novel and original view on the topic.
> >
> > Regarding point 1 made by the authors in the general rebuttal, I find it difficult to imagine raising awareness of algorithmic recourse and engaging with industry and governmental partners through a simple poster session. Probably, a nicer way to attract attention to the shortcomings of algorithmic recourse could have been either propose a tutorial (https://neurips.cc/Conferences/2024/CallForTutorials) or a workshop on the topic (https://neurips.cc/Conferences/2024/CallForWorkshops).
> > Moreover, other surveys on algorithmic recourse and, more broadly, counterfactual explanations (e.g., [1,2]) gained a lot of traction even if not submitted to NeurIPS.
> >
> > [1] Karimi, Amir-Hossein, et al. "A survey of algorithmic recourse: contrastive explanations and consequential recommendations." ACM Computing Surveys 55.5 (2022): 1-29.
> >
> > [2] Verma, Sahil, et al. "Counterfactual explanations and algorithmic recourses for machine learning: A review." ACM Computing Surveys (2020).

---

> ### Author Response · Authors · 2024-08-13
>
> > I am still unconvinced that NeurIPS is the right venue for this contribution, so I will not change my evaluation.
>
> > the points raised by the paper (although all true) are far from giving a novel and original view on the topic
>
> We can see where you are coming from, but we have a different reading of the Call For Papers.
>
> While we cannot infer how the authors of the CFP understand the "new and original research" clause, we still believe that the draft fulfills its requirements. As mentioned in the "global rebuttal" and the response to Reviewer pd5W, the novel aspects of our work include:
>
> 1. evaluating how the *problem* ("tasks") of algorithmic recourse is understood by authors;
> 2. quantifying the insights about this problem, showing, e.g., little interest in looking at AR in the context of specific domains;
> 3. providing a toolkit for other researchers and practitioners to facilitate this systems-oriented outlook on recourse.
>
> We acknowledge and cite in the draft other works that provide various recommendations for AR research that are pertinent (although typically they are challenges for *research* rather than challenges for *research practices*), so we understand if the third point cannot be considered a fully novel contribution of our work.
>
> Nonetheless, to the best of our knowledge points 1. and 2. have not been addressed in the literature before. Given that AR is a broad challenge, we find the lack of its shared understanding, as seen in our results, an important obstacle for the research field.
>
> In our opinion, the "original research" requirement of the CFP is fulfilled due to our choice of the research method: the systematic  (systematized) character of this literature review makes it empirical in nature. It involved data collection, analysis, and interpretation, akin to more common forms of contributions at NeurIPS. Thus, we go beyond "only" summarizing the existing literature, as would generally be the case in non-systematic reviews.
>
> ---
>
> > I find it difficult to imagine raising awareness of algorithmic recourse and engaging with industry and governmental partners through a simple poster session
>
> We agree with you that increasing industry and governmental engagement with algorithmic recourse cannot be achieved by any single publication, but we believe that our work is a good point of departure in that direction. On the one hand, the draft is introductory enough to be valuable for people who are not yet familiar with the field (as also recognized by Reviewer pd5W who commended Section 2 on providing solid background). On the other hand, even researchers who have much experience with research on algorithmic recourse can benefit from a more grounded, shared understanding of the problem.
>
> In any case, we focused our recommendations on key factors that will be relevant for researchers regardless of their exact interest in the field. For instance, the lack of open-source code or documentation will be an important obstacle to the uptake of all forms of algorithmic recourse solutions. We are aware that such recommendations can be perceived as "simple", but they relate to substantial shortcomings that we have identified based on our results.
>
> Following the discussion with Reviewers Wg7R and pd5W we will also make sure that practitioners receive further guidance on how to apply this form of systems-oriented analysis to implement recourse in specific domains.
>
> ---
>
> > Probably, a nicer way to attract attention to the shortcomings of algorithmic recourse could have been either propose a tutorial (...) or a workshop on the topic (...)
>
> We appreciate the suggestion on tutorials or workshops, this is something we will definitely consider as a next step!

---

### Official Review · Reviewer_pd5W · 2024-07-12

**Soundness:** 3
**Presentation:** 3
**Contribution:** 1
**Rating:** 3
**Confidence:** 5

**Summary:**

The paper is a review of algorithmic recourse (AR) literature. The authors deploy a systematic framework to investigate research trends in algorithmic recourse and evaluate their incorporation of practical concerns like societal and institutional considerations of AR, or lack thereof. The review finds that current research is  focused on methods and technical considerations. The authors encourage researchers in AR to consider real-world implications of their work and conduct user studies.

**Strengths:**

- Paper is well-organized and easy to follow
- Section 2 provides solid background information on algorithmic recourse
- The questions in Section 4 are pertinent

**Weaknesses:**

While I agree with the points being made and appreciate the findings in the paper, I question their novelty. As mentioned in Section 4.6, there are papers (albeit in smaller numbers than we would want) that already provide real-world examples and attempt to discuss ethics within recourse. Previous work by [Doshi-Velez and Kim](https://arxiv.org/pdf/1702.08608), [Vaughan and Wallach](https://www.jennwv.com/papers/intel-chapter.pdf) have called for more user studies in interpretable ML, which resulted in studies like [Sixt et al.](https://openreview.net/pdf?id=v6s3HVjPerv). Considering that many researchers working on recourse is also in the field of ML interpretability, I am not sure if the paper's results and call for more user studies are very substantive.

Spending more time differentiating this work from other related works (especially other literature reviews like [70]) rather than listing their contributions in Section 2.2 may be helpful in making your case.

A more thorough discussion and evaluation of results (attempted in Section 5) may resolve some of these questions. As it stands, there is a disconnect between Section 4 and 5. The message of the first part of Section 5 (lines 318-350) is not clear. The second paragraph of the section does not seem to be a discussion of survey results but rather an argument the authors are trying to make (without using the results). The paper would benefit from expanding on the contents in lines 352 to 356, pointing to results in Section 4 and bridging them to the suggestions in Section 5.1.

The paper reads more like a position paper, trying to convince researchers in algorithmic recourse to not only focus on technical methods (which, again, I agree with). But I am not sure if a literature review or a position paper suitable for NeurIPS, considering its call for papers, does not seem to suggest so.

**Questions:**

- What was the motivation for this literature review?
- Why do you say that recourse should be aligned with the *preferences* of the end-users?

**Limitations:**

Yes

---

> ### Author Rebuttal · Authors · 2024-08-06
>
> Thank you for the review and feedback points, we appreciate it!
>
> ---
>
> > As mentioned in Section 4.6, there are papers (albeit in smaller numbers than we would want) that already provide real-world examples and attempt to discuss ethics within recourse.
>
> We believe that the main novel contribution of our work is elsewhere. We recognize that several great works on algorithmic recourse, and/or interpretable ML have called for user studies or considered aspects such as the ethics of AR. In fact, we refer to a few of them in the draft.
>
> Instead, our results point to a broader problem associated with the potential implementation of AR in real-world systems: existing research tends to look at it as a generic mechanism whose implementation does not depend on the context where it would be employed. As one example, even though "actionability" seems to underpin recourse, over 50% of the reviewed publications restrict their definitions of what is actionable to (at best) acting on mutable features. We cannot blame them: this concept is very difficult to define "in the vacuum", without accounting for a specific domain.
>
> We hold fundamental research in high regard. Nonetheless, for algorithmic recourse to become useful, it must be able to address the needs of the systems that it aims to support. Yet, we found few papers that attempt to apply AR. We believe this may stem from the lack of guidance on how it could be operationalized, hence we formulate our recommendations.
>
> ---
>
> > Spending more time differentiating this work from other related works (especially other literature reviews like [70]) rather than listing their contributions in Section 2.2 may be helpful in making your case.
>
> We would be very happy to spend an extra paragraph in the camera-ready version to elaborate on the differences. We will address [70] here, as it was also mentioned by Reviewer Wg7R. If you believe this would strengthen the draft, we are also happy to explain in more depth the differences with other surveys.
>
> The great work by Karimi et al. focuses on the available solutions, while our draft looks at the problem that these solutions aim to address. As the authors note, a major contribution of theirs is a comparison of 60 counterfactual explanation and AR algorithms on technical criteria such as supported models (e.g., neural networks), desiderata (e.g., plausibility), or data types (e.g., tabular). Thus, the analysis in [70] is primarily concerned with publications that propose algorithms; publications with other focus are used in a supporting manner. Moreover, [70] takes an unsystematic approach to reviewing existing publications: while the authors collect an impressive number of documents, it is not clear to what extent they are representative of the field. Their results are also qualitative, rather than quantitative.
>
> Our survey takes a step back compared to [70]: we focus on the understanding of the AR problem and the challenges that authors perceive as necessary to address. Hence, we pose questions about the *characteristics* of research on algorithmic recourse, rather than the *outcomes* of this research. This is also why our analysis is not limited to publications that propose new methods.
>
> ---
>
> > A more thorough discussion and evaluation of results (attempted in Section 5) may resolve some of these questions. As it stands, there is a disconnect between Section 4 and 5.
>
> Again, we agree with you. As explained in the "global rebuttal" we decided to shorten the discussion due to space considerations, but we see your point that in its current shape Sections 4 and 5 may be disconnected. This is something that we can address, as ultimately the entirety of Section 5 follows from our results, even if this is understated in the current draft.
>
> ---
>
> > What was the motivation for this literature review?
>
> Before commencing with this literature review, we had already been relatively familiar with the AR landscape. While we had seen lots of high-quality theoretical work, we were interested in the reasons why it attracts little applied interest. Recourse is a fascinating challenge because it is *necessarily* socio-technical, so we also wanted to evaluate in what ways the social components of recourse are considered in the existing research.
>
> AR is slowly becoming a mature research field, and thus it may impact ADM systems in the future, but as we observe pilot applications remain few and far between, and several challenges may need to be addressed before the industry and governance can benefit from the existing research. Ultimately, we believe that researchers in this domain want to see engagement with the technologies they are creating; our goal was to evaluate how algorithmic recourse is understood by the authors, and through that, why engagement with AR is still missing.
>
> ---
>
> > Why do you say that recourse should be aligned with the _preferences_ of the end-users?
>
> We emphasize the preferences of end-users to address the actionability desideratum of recourse. In many publications, the actionability is assumed to be informed a priori, so we cannot say with certainty where these types of constraints are expected to come from, though many publications mention domain experts as the source of actionability constraints. Nonetheless, we also note that a large majority of authors understand algorithmic recourse as the process of helping end-users overturn undesirable (algorithmic) decisions. Thus, a common understanding of AR seems to be that it is a service to the individual end-users.
>
> For AR to function as a service and give the end-users a real opportunity to overturn undesirable decisions, we see accounting for individual preferences as a crucial aspect of an operational definition. Whether a system can account for these preferences in the form of constraints on features, ranges of features, or something else entirely is a secondary consideration here. Of course, the extent of this alignment is likely domain-dependent.

---

> > ### Comment · Reviewer_pd5W · 2024-08-12
> >
> > Appreciate the detailed answers!
> >
> > Since I slightly missed the main contribution of the paper, it might be worth taking some time to make that clearer in the paper. Perhaps the reason why I understood the main takeaway as calling for more user studies is from the title "grounding and validation of algorithmic recourse in real-world contexts", which one would achieve by running user studies (especially in the context of academic research).
> >
> > Additional things I missed in my first review:
> > - I find the term "socio-technical" vague and unnecessary --- readers reading papers on algorithmic recourse already know its close relation to our lives
> > - The same goes for "society and institutional components". It's an abstract concept that is not easy to grasp (at least for me).
> >
> > Provided that the camera ready version includes an improved discussion section that weaves in its findings, I am willing to adjust my score.
> >
> > Having said that, I am still not confident that the content of the paper is suitable for NeurIPS. As Reviewer yANZ pointed out in their review, the Call for Papers mentions "new and original research" (the paper doesn't fit in the "interdisciplinary" category). I doubt 1) whether the findings are sufficiently novel and 2) contributions are technical enough for NeurIPS, despite their significance in AR research. I will defer this decision to ACs.

---

> ### Author Response · Authors · 2024-08-12
>
> Thank you for getting back to us!
>
> ---
>
> >Since I slightly missed the main contribution of the paper, it might be worth taking some time to make that clearer in the paper.
>
> That is very good feedback. We will outline the contribution of this work in Section 1. Unless you have other suggestions in that regard, we will make use of the (summarized) explanation provided above in our response.
>
> ---
>
> >I find the term "socio-technical" vague and unnecessary --- readers reading papers on algorithmic recourse already know its close relation to our lives
>
> >The same goes for "society and institutional components". It's an abstract concept that is not easy to grasp (at least for me).
>
> We fully agree with you that readers, in general, will understand that recourse would have impacts on people's lives. We talk about these social and institutional dimensions of algorithmic recourse problem in a somewhat broader sense, similar to how they are understood in fields such as systems safety, e.g.:
> * Leveson, N. G. (2012). _Engineering a Safer World: Systems Thinking Applied to Safety_ (p. 69). The MIT Press.
> * De Bruijn, H., & Herder, P. M. (2009). System and actor perspectives on sociotechnical systems. _IEEE Transactions on systems, man, and cybernetics-part A: Systems and Humans_, _39_(5), 981-992.
>
> With this, we aim to emphasize that the understanding of recourse in a specific domain will be influenced by factors such as the involved stakeholders or the organizational processes (see, e.g., lines 377-379), besides the technical factors that are the focus of the existing body of research. Moreover, many problems associated with algorithmic recourse, such as the meaning of actionability or its propensity to lead to unexpected emergent dynamics, can be understood only when accounting for all these components. In any case, we are happy to provide explicit definitions for these terms in the Introduction for ease of exposition.
>
> ---
>
> > Provided that the camera ready version includes an improved discussion section that weaves in its findings, I am willing to adjust my score.
>
> Of course! To recap, we will reorganize Section 5 so that it starts with a longer discussion of the results, including what is now lines 352-356 to better explain the provenance of our five recommendations. Next, we will associate with each recommendation the guidelines for practitioners (from our answer to Reviewer Wg7R), explaining how these can be implemented in specific domains. Finally, we will integrate the current lines 318-350 as they follow from the recommendations, reducing the "disconnect" that you have highlighted in your initial review.
>
> We are confident that these changes can be duly implemented within ≈three-fourths of the additional page in the camera-ready version of the draft, leaving enough space to better differentiate our work and the earlier reviews, as we also discussed.
>
> ---
>
> > I doubt 1) whether the findings are sufficiently novel and 2) contributions are technical enough for NeurIPS, despite their significance in AR research. I will defer this decision to ACs.
>
> We respect your judgment and appreciate your deferral of this decision to the ACs.
>
> ---
>
> *Edit:* We have not received any notification from OpenReview that the above comment was received, so we are updating it to ensure that a notification was sent to the Reviewer. There are no changes in our response compared to its earlier version.

---

### Official Review · Reviewer_bnWe · 2024-07-13

**Soundness:** 3
**Presentation:** 2
**Contribution:** 2
**Rating:** 7
**Confidence:** 4

**Summary:**

The authors present a survey regarding algorithmic recourse scientific literature. In their work, the authors analyze what types of contributions do the authors choose to make to the AR research, what are the criteria covered in the authors’ definitions of AR, what are the criteria covered in the authors’ definitions of actionability, the roles of end users, what types of real-world considerations motivate existing research, what types of real-world considerations are seen as challenges for future work, what types of group-level dynamics are addressed in the existing research, what are the approaches to the realistic evaluation of proposed methods, and what are the open source and documentation practices in AR research. They conclude their paper by providing recommendations on how to make future algorithmic recourse solutions better suited for real-world needs.

**Strengths:**

- the authors invested much effort into explaining the procedure followed to ensure a high-quality survey
- the authors very synthetically review scientific literature related to algorithmic recourse and provide a great insight into the field within a few pages
- the authors reviewed a vast amount of literature (165 references!)

**Weaknesses:**

We did not identify important weaknesses. While an extensive survey could be created following this one, providing in-depth details for each of the sections, we understand this cannot be done within the constraints established for this venue.

**Questions:**

We consider the work to be interesting and relevant. We consider the review to be concise yet relevant, and that could be extended later, providing more fine-grained insights on the topic of algorithmic recourse. Nevertheless, we would like to point to the following improvement opportunities:

(1) - the authors in their abstract mention multiple times the actionable component of algorithmic recourse, which is present in some of the subsections, but in others, the link to this aspect is less clear and could be enhanced.

(2) - did the authors find any works considering actionability in a wider sense, e.g., that some action could be taken even by machines? What are the implications and concerns in such cases?

**Limitations:**

The authors have adequately acknowledged the limitations of their work.

---

> ### Author Rebuttal · Authors · 2024-08-06
>
> Thank you very much for your positive evaluation of our paper!
>
> ---
>
> > (1) - the authors in their abstract mention multiple times the actionable component of algorithmic recourse, which is present in some of the subsections, but in others, the link to this aspect is less clear and could be enhanced.
>
> We are happy to enhance this connection wherever possible. While reviewing the existing literature, we have observed that actionability is understood as the primary consideration for algorithmic recourse, distinguishing it from "passive" explanations. In the current draft, we emphasized the actionability considerations when they influenced the contributions of other authors. For instance, authors who understand actionability as *"acting on mutable features"* tend to incorporate this requirement into their AR solutions, but not more (e.g., not an interface that would allow the end-user to specify their preferences).
>
> ---
>
> > (2) - did the authors find any works considering actionability in a wider sense, e.g., that some action could be taken even by machines? What are the implications and concerns in such cases?
>
> This is a very interesting question, especially since, as we note in the draft, the understanding of actionability tends to be relatively limited in the existing research. We have not identified any works where an automated system would take actions on behalf of the end-users (e.g., a machine canceling customer's lines of credit to improve their credit score). Nonetheless, the considerations of Venkatasubramanian and Alfano [142] who proposed that in some settings algorithmic recourse may require (human) fiduciaries, or even Slack et al. [125] who considered that model owners are not guaranteed to be trustworthy, seem to apply here. Ultimately, this is a question of control and accountability, e.g.:
> * What types of decisions can an algorithm take on behalf of an end user?
> * Who is responsible if the automated action leads to an adverse outcome?
>
> Also, certain design decisions in recourse solutions inherently involve broadening or restricting the actionability of the generated recommendations on behalf of the user. For instance, many authors postulate that generating diverse recommendations improves their actionability, but we have also noted dissenting voices such as Albini et al. [6] who suggest that diversity may lead to cognitive overload for the end-users. Finally, we note that some works attempt to automatically discover what is actionable, e.g., the work of Kelechi and Jiao [72] on quantifying actionability. That line of work is relatively unexplored.

---

> > ### Comment · Reviewer_bnWe · 2024-08-12
> >
> > Thank you for the comments. We acknowledge we have reviewed the responses and have no further comments.

---

> ### Author Response · Authors · 2024-08-13
>
> We also acknowledge having seen the response of Reviewer bnWe and thank them for the confirmation.

---

### Author Rebuttal · Authors · 2024-08-06

Dear Reviewers,

First and foremost, we would like to thank you for your insightful and comprehensive comments. We know that the review process tends to be time-consuming, and we are grateful that you took this time to read our paper in depth and produce reviews of such high quality.

We are also delighted that — regardless of the preliminary scores — your reviews commend the quality of our work and the relevance of the literature review that we have undertaken.

We address the specific points you have raised in the individual responses. In this "global rebuttal", we instead want to focus on the two main objections that seem to have resulted in the three rejections. More specifically, we:

1. explain why we have decided to submit this work to NeurIPS and still believe that it is suitable for the venue; and

2. outline how we can address the suggestion of Reviewer Wg7R that our draft is not achieving its full potential within the additional one page of content in the camera-ready version, should you decide to accept our paper.

-----

**Regarding point 1.**, we agree with the comments that a literature review is not a typical contribution for NeurIPS. We decided to submit this work to NeurIPS, because we believe that algorithmic recourse could become a valuable safety mechanism in algorithmic decision-making systems, provided that it attracts interest from industry and governmental partners. NeurIPS actively engages with these partners and, as also recognized by Reviewer Wg7R, they form an integral part of its audience. In our literature review, we have observed that research on algorithmic recourse remains driven by academia with relatively little interest from other communities to pilot such mechanisms. In our opinion, this work can help bridge the gap and increase engagement with (existing) research on AR. We are very happy that the potential of this work was also recognized by the Reviewers.

Furthermore, we believe that our work complies with the Call for Papers in that NeurIPS invites papers on *"social and economic aspects of machine learning"* and also *"interdisciplinary submissions that do not fit neatly into existing categories"*. While we acknowledge that other authors have previously pointed out the lack of user studies in interpretable ML (as noted by Reviewers pd5W and Wg7R) , we reckon that reinforcing this sentiment is only a part of our work.

Most of all, our contribution is empirical in nature in that we review the landscape of algorithmic recourse and *quantify* the insights underlying existing work. We are able to put numbers to these insights because — differently from the previous reviews in the field — we follow a systematized methodology. Based on the empirical results, we explain that for algorithmic recourse to satisfy the socio-technical requirements of systems where such mechanisms would be applied, future research requires a broader scope: not only by involving users but also, crucially, by looking at AR solutions in the context of specific real-world domains.

-----

**Regarding point 2.**, due to the nine content pages limit we decided to reduce the length of Section 5 (Discussion) to better introduce the results in Section 4. We believe that with the additional page of content in the camera-ready version, we will be able to properly address the well-grounded concerns of the Reviewers.

In particular, we decided to only briefly explain the practical aspects of introducing algorithmic recourse into specific domains in lines 338-350, where we discuss the stark differences in what AR would entail in two contexts: education and medicine. This analysis highlights that several important questions for AR research cannot be answered without attending to its operational aspects. For example, in Section 4.3 we have looked at the definitions of "actionability" and noted that while this concept is crucial for algorithmic recourse (frequently equated with *"actionable counterfactual explanations"*), its understanding is limited.

In our opinion, actionability can only be understood for a *specific* problem: domain, application, context, stakeholders, etc. As we explain, in a setting such as providing recommendations to improve learning outcomes almost any suggestion may be acted upon by an affected student. Meanwhile, in a more involved setting such as attempting to improve health outcomes, there will exist a variety of additional constraints on the recommendations (e.g., availability of medications) and the involved stakeholders (e.g., a clinician implementing recourse on behalf of the patient). Moreover, algorithmic recourse may even be altogether impossible if improving the health outcomes of a patient is beyond the capabilities of medicine.

We understand that this analysis is understated in the current draft and we will be happy to elaborate on the steps that can be taken to put our recommendations into practice in specific real-world contexts (see also our answer to the comments of Reviewer Wg7R). This way we can also address the feedback of reviewer pd5W who noted that it is currently not clear how the first part of Section 5 ties to the results in Section 4. The five recommendations in Section 5.1 directly follow from the main "problems" we have observed in the literature review (we very briefly note this in lines 352-356), and we will be able to explain in more detail how they can be put into practice in a specific real-world context with the additional page of content. In our opinion, this requires only a minor revision of the paper.

---

We are looking forward to reading your further comments,
All the best,
The Authors

---

> ### Author Response · Authors · 2024-08-13
> **All Official Comments are answered**
>
> Dear Reviewers,
>
> We have now answered all four Official Comments submitted by you during the discussion period.
>
> We will keep monitoring OpenReview in case you have any last-minute clarification questions, and do our best to properly address them in time.
>
> Once again we would like to thank you for the productive discussions,
> All the best,
> The Authors

---

### Decision · Program_Chairs · 2024-09-25

**Decision:**

Reject

**Comment:**

The paper conducts a thorough review of the existing literature on algorithmic recourse, focusing on the disconnect between theoretical approaches and practical applications in real-world settings. The authors review 127 publications and identify key themes, challenges, and gaps in the current research, emphasizing the need for more grounded and practical considerations in the development of algorithmic recourse methods. The paper concludes with five recommendations aimed at bridging the gap between research and practical implementation.

The reviewers generally agree that the paper is well-written, comprehensive, and provides valuable insights into the current state of algorithmic recourse research. There is a strong consensus among the reviewers that this is a good survey paper.

The main decision, therefore, depends on whether NeurIPS is the right venue for it. After confirming with the SAC, the call for papers requires submissions to contain "new and original research," and unfortunately, this paper does not satisfy that requirement. I tentatively recommend rejection on this ground but will leave a note for the SAC/PC to make the final decision. If the paper is rejected, reviewer yANZ has provided a list of pointers on where this submission might be better suited, and we strongly encourage the authors to incorporate the reviewer comments and submit the work to another venue.